# Analytical Pyrolysis of *Pinus radiata* and *Eucalyptus globulus*: Effects of Microwave Pretreatment on Pyrolytic Vapours Composition

**DOI:** 10.3390/polym15183790

**Published:** 2023-09-17

**Authors:** Diego Venegas-Vásconez, Luis E. Arteaga-Pérez, María Graciela Aguayo, Romina Romero-Carrillo, Víctor H. Guerrero, Luis Tipanluisa-Sarchi, Serguei Alejandro-Martín

**Affiliations:** 1Departamento de Ingeniería de Maderas, Universidad del Bío-Bío, Concepción 4081112, Chile; diego.venegas1801@alumnos.ubiobio.cl (D.V.-V.); larteaga@ubiobio.cl (L.E.A.-P.); maguayo@ubiobio.cl (M.G.A.); 2Laboratorio de Cromatografía Gaseosa y Pirólisis Analítica, Universiad del Bío-Bío, Concepción 4081112, Chile; 3Laboratorio de Procesos Térmicos y Catalíticos, Universidad del Bío-Bío, Concepción 4081112, Chile; 4Centro de Biomateriales y Nanotecnología, Universidad del Bío-Bío, Concepción 4081112, Chile; 5Departamento de Química Analítica e Inorgánica, Facultad de Ciencias Químicas, Universidad de Concepción, Concepción 4070371, Chile; rominaromero@udec.cl; 6Departamento de Materiales, Escuela Politécnica Nacional, Quito 170525, Ecuador; victor.guerrero@epn.edu.ec; 7Facultad de Mecánica, Escuela Superior Politécnica de Chimborazo, Riobamba 060155, Ecuador; luis.tipanluisa@espoch.edu.ec

**Keywords:** biomass, pyrolysis, pretreatment, microwave, oxygenated compounds

## Abstract

*Pinus radiata* (PR) and *Eucalyptus globulus* (EG) are the most planted species in Chile. This research aims to evaluate the pyrolysis behaviour of PR and EG from the Bío Bío region in Chile. Biomass samples were subjected to microwave pretreatment considering power (259, 462, 595, and 700 W) and time (1, 2, 3, and 5 min). The maximum temperature reached was 147.69 °C for PR and 130.71 °C for EG in the 700 W-5 min condition, which caused the rearrangement of the cellulose crystalline chains through vibration and an increase in the internal energy of the biomass and the decomposition of lignin due to reaching its glass transition temperature. Thermogravimetric analysis revealed an activation energy (E_a_) reduction from 201.71 to 174.91 kJ·mol^−1^ in PR and from 174.80 to 158.51 kJ·mol^−1^ in EG, compared to the untreated condition (WOT) for the 700 W-5 min condition, which indicates that microwave pretreatment improves the activity of the components and the decomposition of structural compounds for subsequent pyrolysis. Functional groups were identified by Fourier transform infrared spectroscopy (FTIR). A decrease in oxygenated compounds such as acids (from 21.97 to 17.34% w·w^−1^ and from 27.72 to 24.13% w·w^−1^) and phenols (from 34.41 to 31.95% w·w^−1^ and from 21.73 to 20.24% w·w^−1^) in PR and EG, respectively, was observed in comparison to the WOT for the 700 W-5 min condition, after analytical pyrolysis. Such results demonstrate the positive influence of the pretreatment on the reduction in oxygenated compounds obtained from biomass pyrolysis.

## 1. Introduction

Population growth, urbanisation, and industrialisation have increased fossil resource consumption or energy and chemical production [1]. Currently, 12% of the world’s oil is dedicated to chemical production, which serves as an essential precursor for high-value-added products such as fertilisers, plastics, rubber, fibres, and solvents [2]. Nevertheless, the intensive use of fossil resources has given rise to environmental issues such as global warming and climate change. Given this situation, transitioning from this fossil-based economic model to alternative bio-based economies is essential [1]. Biomass can play a crucial role in such a transition due to its availability, compatibility with existing energy infrastructure, and versatility in various applications. Accordingly, several studies have been reported using agricultural and forestry residues, livestock manure, municipal waste, organic sewage, and industrial wastes to replace oil in its current applications [3,4]. Among them, lignocellulosic biomass (LCB), with an annual production rate of over 180 billion tons, stands out as a promising source [2].

Lignocellulosic biomass is a complex material composed of cellulose, hemicellulose, lignin, and extractives [5]. Cellulose is a homopolymer made of glucose units, while hemicellulose is a polysaccharide made of pentoses, hexoses, and organic acids. Lignin is an aromatic polymer made of phenylpropane units that provides structural rigidity. Extractives are chemical compounds in biomass that contribute to their mechanical properties, biological resistance, and quality despite their small percentage of the total composition [6].

Chile has significant commercial plantations of lignocellulosic biomass. According to the Forestry Yearbook of the Chilean Forestry Institute—INFOR [7], in 2020, the country had a forest area of 17 million ha, which represented 22.5% of the total national area, of which 14.7 million ha correspond to native forests and 2.3 million ha to forest plantations, where *Pinus radiata* and *Eucalyptus* spp., with 92.66%, were the most planted species, with yields ranging from 25 to 35 m^3^·ha^−1^·year^−1^. Additionally, silvicultural activity and the processing of forest resources generate over 4 million metric tons of energetically exploitable residues [8].

Pyrolysis is one of the thermochemical conversion technologies which can transform biomass into valuable products. This process takes place over a wide range of temperatures (usually between 450 °C and 550 °C), converting the biomass into three significant fractions: biochar, a potential adsorption material, a catalyst, soil amending material, etc.; non-condensable gases, for supplying the pyrolysis’s thermal requirements; and the pyrolysis oil, which could be used as a fuel or as a platform for the petrochemical industry [6]. The pyrolysis oils are a complex mixture of organic compounds from different chemical families, such as acids, alcohols, aldehydes, esters, ketones, and phenols. Despite being a promising alternative energy source, pyrolysis oil faces technical difficulties as compared to conventional fuels (i.e., diesel); for example, its high acidity (pH 2–3), high viscosity (40–100 cP), high water content (25–40%), low heating value (16–19 MJ.kg^−1^), and corrosiveness [9]. Several studies have addressed the limitations of pyrolysis oils by using pretreatments or by upgrading processes downstream of the pyrolysis reactor [9,10,11,12,13].

Common pretreatments focus on reducing the biomass water content by drying and torrefaction, yielding pyrolysis oils with reduced oxygen contents [14]. Moreover, solvolytic transformations of the biomass (i.e., lignin-first or cellulose-first approaches) can produce a preliminary degradation of the biomass, thus favouring the selectivity to specific functional groups (aromatics, phenolics, or sugars) [15]. Biomass microwave (MW) processing has recently emerged as a promising pretreatment because it induces autohydrolysis, separating hemicellulose and lignin from cellulose [16]. In addition, MW features several advantages when compared to traditional conduction- or convection-based treatments. Microwave irradiation induces the alignment of polar molecules, causing their dipole moments to align with the radiation field, resulting in polarisation. This realignment of polar molecules generates heat by causing displacement within the material [17]. The frequency of 2.4 GHz is commonly used in microwave production, enabling uniform and efficient heating with an energy transformation efficiency of approximately 80%. Microwaves have a relatively low photon energy (0.03 kcal·mol^−1^), which does not directly impact molecular structures since the chemical bond energies range from 20 to 50 kcal·mol^−1^. Therefore, microwaves, as a form of non-ionising radiation, can transform biomass without affecting structural bonds in its polymeric structure [18].

Wang et al. [19] compared microwave and traditional oven-drying at 105 °C as pretreatments for pyrolysis, using four power levels (200, 400, 600, and 800 W) on pine sawdust, peanut shells, and corn stalks and inspected the product distribution from the fast pyrolysis process. There was little change in the biochar yield, and the bio-oil yield increased, while the yield of non-condensable gas decreased by microwave compared to those obtained after oven pretreatment. In addition, MW pretreatment reduced the water content in the pyrolysis oil from 26.8% to 26.2%, while the viscosity remained unaltered and the lower heating value increased from 14.72 to 14.97 MJ·kg^−1^. Contrary to the physical properties, the MW was quite relevant in affecting the pyrolysis product distribution, confirmed by a decrease in oxygenated compounds such as acetol from 31.54% to 11.78% and 2-methoxyphenol from 1.98% to 1.02%, respectively. The minimal impact on the physical properties, and the relevant role of pyrolysis’ product distribution of microwave pretreatment, demonstrated in Wang’s work [19], is inconclusive and requires further analysis to unravel how the structural and chemical effects of microwaves correlate with the pyrolysis reaction routes. Gao et al. [20] partially addressed this question and applied three levels of microwave power (380 W, 540 W, and 700 W) and a 1 min dwell time to tobacco stalks before pyrolysis. When compared to pyrolysis oils obtained from the same biomass but using traditional oven pretreatments, the content of alcohols, acids, aldehydes, and ketones from MW-treated tobacco stalks increased to 6.46%, 3.39%, 18.59%, and 18.18%, respectively. In comparison, the relative content of phenols decreased to 8.74%.

Despite the extensive literature reporting on the effect of microwaves as a pretreatment of lignocellulosic biomass, there is currently no consensus on their influence on pyrolysis vapours. Prior evidence indicates that microwaves reduce the moisture content of the biomass and alter its chemical composition (cellulose, hemicellulose, lignin, and extractives), which can influence the pyrolysis vapour composition. Therefore, the authors use PR and EG biomasses to study humidity loss and chemical and physical composition, focusing on investigating the impact of microwave pretreatment on pyrolysis vapours.

## 2. Materials and Methods

### 2.1. Raw Material

The Investigaciones Forestales Bioforest S.A. company from Bío Bío, Chile, provided biomass samples (PR and EG) as chips. The chips were exclusively made of wood without bark, and the position of the sample within the trunk was not considered (sapwood/heartwood). Samples were ground and sieved to achieve an average particle size of 0.33 mm and stored in a desiccator until further use, according to the procedure reported elsewhere [21].

### 2.2. Methods

#### 2.2.1. Biomass before Pretreatment Characterisation

##### Proximate, Ultimate Analysis and Calorific Value

The elemental analyses for PR and EG samples were performed using a Leco CHNS 628 elemental analyser (LECO corporation, St. Joseph, MI, USA) according to the ASTM D5373 Standard method [22]. The proximate analyses for moisture content, volatile matter, ash content, and fixed carbon were performed in a muffle furnace using the ASTM D3172 Standard method [23]. The oxygen (O) was calculated by the difference of carbon (C), hydrogen (H), and nitrogen (N).

The high heating value [MJ·kg^−1^] was estimated by Channiwala’s correlation [24] for solid, liquid, and gaseous fuels, according to Equation (1):(1)$$HHV=0.3491.C\left(\%\right)+1.1783.H\left(\%\right)+0.1005.S\left(\%\right)-0.1034.O\left(\%\right)\;-0.0151.N\left(\%\right)-0.0211.Ash(\%)$$

##### Chemical Composition

A typical analysis used about 300–400 mg of PR and EG with 45–60 mesh to quantify the extractive content. The procedure is conducted in a Soxhlet system using acetone:water (9:1) solution, and performing the extraction for 16 h was recommended by the TAPPI T280 pm-99 (2000) standard [25]. Holocellulose and alpha-cellulose contents were measured according to TAPPI T9 wd-75 (2015) [26]. Holocellulose was isolated from 250 mg of extractive-free sample by adding 5 mL of deionised water, 2 mL CH_3_COOH, and 5 mL of 80% NaClO_2_; the samples were immersed in a water bath at 90 °C for one hour. Then, 4 mL of CH_3_COOH and 10 mL of 80% NaClO_2_ were added to the flask, and the reaction continued for one hour at 90 °C. The suspension was filtered, washed with deionised water, and dried in an oven at 105 °C until it reached a constant weight. From 100 mg of holocellulose samples, alpha-cellulose was obtained by reaction with 8 mL of NaOH (17.5%) for 30 min at room temperature with stirring every 10 min. Later, 8 mL of deionised water was added to the solution, and the reaction was continued for 30 min. The solids were filtered and washed with deionised water, and the yield of alpha-cellulose was determined by weight. The lignin content was determined using extractive-free samples according to TAPPI T222 om-11 (2015) [27]. The sample (300 mg) was hydrolysed with 3 mL H_2_SO_4_ (72% w·w^−1^) at 30 °C for 1 h; later, the acid was diluted to 4% (w·w^−1^) using distilled water. The mixture was transferred to a 250 mL Erlenmeyer flask and autoclaved for 1 h at 121 °C; subsequently, the residual material was cooled and filtered, and the solids were dried to a constant weight at 105 °C and classified as insoluble lignin. Soluble lignin was determined by measuring the absorbance of the solution at 205 nm [28]. The total lignin content was calculated from insoluble and soluble lignin. All analyses were carried out in triplicate.

#### 2.2.2. Microwave Irradiation

The microwave treatments were carried out at four power levels (259, 462, 595, 700 W) and irradiation times (1, 2, 3, 5 min) in a Thomas TH-18B03 microwave oven (230 V, 50 Hz, operating frequency 2450 MHz). Each trial was conducted three times for repeatability, reporting the average values. The average (T_avg_) and maximum temperatures (T_max_) within the microwave oven were recorded using an Optris PI160 infrared camera (320 × 240 pixels) (Optris, Berlin, Germany), and the data analysis was performed with the PI Connect GEN-E2014-02-A software, according to a procedure reported elsewhere [29]. The biomass samples’ water loss was calculated using Equation (2) after varying the microwave pretreatment conditions. The initial water content was measured on an XY200MW thermobalance.
(2)$$W_{loss}=\frac{W_{in}-W_{fin}}{W_{fin}}$$
where W_loss_ is the water percentage loss of the biomass sample, W_in_ is the initial sample weight, and W_fin_ is the sample weight after pretreatment.

The primary energy consumed during microwave pretreatment under different conditions (microwave operating power and duration dwell time) is shown in Equation (3).
(3)Q=P ∗ t
where Q is the primary energy required in the process (J), P is the microwave operating power (W), and t is the microwave application time (s).

#### 2.2.3. Biomass Pretreated Characterisation

##### Thermogravimetric Analysis (TGA)

TGA experiments were performed on a TGA Q500 thermobalance (TA Instruments, New Castle, DE, USA). Approximately 10 ± 1 mg of sample was heated at (5, 10, and 20 °C min^−1^) from room temperature to 650 °C. The experiments were conducted at atmospheric pressure with an N_2_ flow (50 mL·min^−1^).

Non-isothermal thermogravimetric analysis examines the mass change as a function of temperature. Conversion (α) is defined by Equation (4) [30]:(4)$$\alpha (t)=\frac{m_{i}-m_{t}}{m_{i}-m_{f}}$$
where m_i_ is the initial sample mass, m_t_ is the sample mass at time t, and m_f_ is the final sample mass.

Such data can be kinetically interpreted through Equation (5) [30]:(5)$$\frac{d\alpha }{{dt}_{\alpha }}=k\left(T\right)f\left(\alpha \right)=A\;exp\left(-\frac{E_{a}}{RT_{\alpha }}\right)f(\alpha )$$
where dα/dt_α_ expresses the conversion rate as a function of time, k is the rate constant and is a function of temperature, f(α) is a function of the reaction mechanism, A is the pre-exponential factor (min^−1^), E_a_ is the apparent activation energy (kJ·mol^−1^), R is the universal gas constant (8.314 J K^−1^·mol^−1^), and T is the absolute temperature (K).

As biomass pyrolysis is a multi-step process involving several reactions, one plausible approach to unravelling its kinetic behaviour is applying isoconversional methods. Numerous papers demonstrate that Friedman’s differential method, Flynn–Wall–Ozawa (FWO), and Coats–Redfern (CR) models are suitable for interpreting TGA data from biomass pyrolysis. Therefore, these approaches (Table 1) are adopted here to understand the possible kinetic implications of the MW pretreatment of biomass.

##### Fourier Transform Infrared (FTIR) Spectroscopy

Mid-infrared spectra of untreated and pretreated biomass samples were collected using a Thermo Spectrometer Scientific Infrared Nicolet iS10 with a DTGS detector, equipped with a Quest ATR accessory (Specac, Orpington, UK). The spectra were averaged from 32 scans using a resolution of 4 cm^−1^ over the mid-IR range (4000–400 cm^−1^). The crystallinity index (CI) of cellulose was determined from the FTIR spectra to verify to the extent to which the MW affected the physical structure of the biomass (Equation (6)) [35]:(6)$$CI=\frac{A_{1429}}{A_{897}}$$
where A_1429_ is the absorption band intensity for cellulose (I) at 1429 cm^−1^, and A_897_ is the absorption band intensity for cellulose (II) at 897 cm^−1^.

##### Morphology Analysis

Changes in biomass surface resulting from MWI pretreatment were analysed in a JEOL JSM-6610LV (JEOL, Tokyo, Japan) scanning electron microscope (SEM) operating at 5 kV and with 500× magnification.

#### 2.2.4. Extractive Characterisation

The extractive compounds were analysed by gas chromatography (GC, 6890 N GC, Agilent Technologies, Santa Clara, CA, USA) equipped with a mass spectrometry detector (5975C MSD, Agilent Technologies). Before the analysis, extractives were derivatised by silylation (BSTFA + TMCS); using a mixture of 100 μL of pyridine, 0.5 mL of hexamethyldisilazane (HDMS), and 0.3 mL of trimethylchlorosilane (TMCS) as a solvent. According to Fabbri et al., the mixture was heated in a quartz tube under N_2_ flow for 30 min at 60 °C [36].

#### 2.2.5. Pyrolysis Analysis

The biomass fast pyrolysis was carried out in an analytical micro-pyrolysis reactor (Pyroprobe 5200HPR, CDS Analytical Co., Ltd., Oxford, PA, USA) connected in line with a gas chromatograph equipped with a mass spectrometer system, GC/MS (Clarus 690, QS8. Perkin Elmer, Waltham, MA, USA). The pyrolysis reactor consisted of a quartz tube heated by a platinum filament. Approximately 0.5 ± 0.1 mg of each sample was accurately weighed by a microbalance (AD 6000 Ultra MicroBalance Perkin Elmer). According to previous analyses, the experiments were conducted at 550 °C to guarantee maximum biomass decomposition and to achieve the highest pyrolysis oil yield [5]. He (pure 99.996%, BOC ICONSA, Concepción, Chile) was used as a carrier gas. Compound identification was performed using the NIST 2017 library and TurboMass 6.1.0 software. The pyrolysis experiments were analysed for biomass samples with extractables (WE) and without extractables (WOE).

## 3. Results

### 3.1. Biomass Untreated Characterisation

#### Proximate, Ultimate Analysis, Calorific Value, and Chemical Composition

Table 2 shows proximate and ultimate analyses, the higher heating value, and the raw material chemical composition (PR and EG). Proximate analysis showed no significant differences among these wood species, featuring a high volatile content that validates their pyrolysis feasibility. Notably, the ash content of these samples was below 1%, allowing the discarding of any catalytic effect of ashes during pyrolysis [37].

The ultimate analysis poses the PR and EG at the same level as other woody biomass previously reported in the literature [38]. The high carbon (47–48%) and low N_2_ (0.09–0.30) contents also confirm the suitability of PR and EG for thermochemical transformation processes and suggest that this raw material is unlikely to generate NOX and SO_X_.

The results obtained for chemical composition agree with those published by Wang et al. [39]. For softwoods (e.g., pine), the cellulose content ranges from 40 to 44%, along with hemicellulose (25–29%), lignin (25–31%), and extractives (1–5%). For hardwoods (e.g., *Eucalyptus*), the cellulose content ranges from 43 to 47%, alongside hemicellulose (25–35%), lignin (16–24%), and extractives (2–8%).

### 3.2. Microwave Pretreatment

The upper limit for microwave operation was fixed after preliminary analyses, which confirmed that at 700 W and exceeding 5 min, the biomass undergoes self-incineration. Therefore, the study considered a maximum pretreatment condition at 700 W for 5 min (210,000 J) and three intermediate points at 259 W, 462 W, and 595 W. Each trial was tested three times for repeatability, and the average value was reported.

Figure 1a,b show the average (T_avg_) and maximum (T_max_) temperatures reached by the biomass samples vs. power, and Figure 1c,d show weight loss (W_loss_) vs. power resulting from microwave treatment for PR and EG, respectively. Additionally, Appendix A summarises all of the microwave pretreatment parameters for PR and EG. In the microwave pretreatment condition of 700 W-5 min, the highest mean temperature (T_avg_) (115.17 °C and 101.98 °C) and the highest maximum temperature (147.69 °C and 130.71 °C) were reached for PR and EG, respectively. The initial moisture content in PR (7.75%) and EG (5.51%) contributed to an increase in the system’s internal kinetic energy through the vibration produced by the microwave in the water molecules [18], which contributed to a higher T_max_ in PR compared to EG. With these maximum temperatures, it is suggested that during the pretreatment, new extractives were generated between 120 and 250 °C [40]. The 700 W-5 min condition produced the highest water loss (W_loss_) for both PR (11.06%) and EG (5.17%). In this condition, the highest energy is delivered to the biomass, which, when converted into heat, causes water evaporation [41]. Similar results were achieved by Fia and Amorim [17], confirming that a higher power and a longer irradiation time lead to higher temperatures and higher water removal from biomass.

The temperature variations between the 259 and 700 W for 1, 2, 3, and 5 min are 25.94, 24.43, 38.02, and 41.82 °C for PR and 29.59, 42.42, 48.84, and 47.26 °C for EG, respectively. It is observed that as the irradiation time increases, the temperature variations also increase, and in the specified conditions (259 and 700 W), these variations are the largest compared to the variations in the intermediate treatments (462 and 595 W). Therefore, extreme microwave conditions (259 and 700 W) and 5 min were chosen to evaluate the most significant changes in subsequent pyrolysis experiments.

### 3.3. Effect of MW on Biomass Chemical Composition

Table 3 shows the variation in the lignocellulosic species’ chemical composition after the microwave treatment. In all cases, the difference between the standard deviation and the arithmetic mean is less than 6%, except for the extractives in the 700 W-5 min condition, which reaches 10.52%. Such variations in chemical components are due to the pretreatment effect and the microwaves acting inside the biomass and cannot be significantly attributed to experimental measurement error. The interactions between electromagnetic waves and matter are expressed as conductive losses, dielectric losses, and magnetic losses [42]. The alignment of polar molecules caused by microwave irradiation forces the dipoles of the molecules to align in the radiation field (polarised). The rearrangement of the polar molecules produces a displacement within the material that generates heat [43].

In the maximum condition (700 W-5 min), the cellulose content decreased after pretreatment from 43.1% to 41.3% in PR and from 53.0% to 50.4% in EG. Such variation can be associated with microwave irradiation that leads to intense molecular collisions due to dielectric polarisation, potentially decomposing the cellulose content [44]. Then, the cellulose structure’s hydroxyl groups (-OH) result in a polar molecule. Then, as the microwaves pass through these polar molecules, they realign and vibrate them, increasing their internal temperature and displacement [18]. In this condition, the lignin content decreased after pretreatment, from 26.6% to 24.4% in PR and from 23.9% to 22.3% in EG. The glass transition temperature of lignin (130–200 °C) is usually lower than that of cellulose (230–250 °C) and close to hemicellulose (160–200 °C) under a dry state [45]; therefore, the decrease in this component may be associated with the initiation of decomposition due to reaching its glass transition temperature, which results in the disorder of the lignin structure.

Additionally, under these circumstances, new extractives can be generated when the wood is thermally treated due to the degradation of other components, such as lignin [46,47]. This generation of new extractives is reflected in its increase from 1.8% to 7.2% in PR and from 1.9% to 6.7% in EG. Finally, the arousal effect of the molecules by microwave application only increases the internal kinetic energy of the biomass. Furthermore, since microwaves have a low energy in their photons (0.03 kcal/mol), they do not directly affect the molecular structures of the biomass, as their chemical bonds have an energy ranging from 20 to 80 kcal/mol, resulting in a general percentage increase in hemicellulose content [48].

### 3.4. Biomass Pretreated Characterisation

#### 3.4.1. Thermal and Kinetic Parameters

##### Thermal Decomposition Characteristics and Heating Rate Effects

Figure 2a,b show the thermogravimetric (TG), differential thermogravimetric (DTG), and second derivative of mass loss (−d^2^m/dt^2^), also known as D^2^TG for PR and EG at 10 °C·min^−1^, respectively. The Savitzky–Golay method smooths the TG and DTG curves obtained from TGA assays [49]. The initial degradation temperature (T_initial_) is assumed to correspond to a solid mass fraction equal to 0.975 (not shown). The T_shoulder_ corresponds to the temperature of the maximum decomposition rate of hemicellulose, and the overall maximum of the hemicellulose decomposition rate is dY/dT_shoulder_. The temperature T_peak_, where the maximum devolatilisation rate is attained (associated mainly with cellulose decomposition), is introduced with the corresponding −(dY/dt)_peak_ and Y_peak_. T_onset_ and T_offset_ are the boundaries between the three characteristic zones: (1) moisture loss and extremely volatile components (extracts), (2) active pyrolysis, and (3) passive pyrolysis, where mainly biochar is formed. At the end of the thermogravimetric test (650 °C), there is a residue percentage (% Res) corresponding to solid residue production [50].

Appendix A summarises the devolatilisation characteristics obtained from the curves shown in Figure 1. The temperature values (T_initial_, T_onset_, T_shoulder_, T_peak_, and T_offset_) increase as the microwave irradiation increases from the without-treatment (WOT) samples to the samples irradiated at 700 W-5 min for PR and EG. These results are consistent with the results presented by Grønli et al. [51] in their work on thermogravimetric analysis and the devolatilisation kinetics of hardwoods (beech, alder, birch, and oak) and softwoods (Douglas fir, pine, redwood, and spruce), whose average values were: T_onset_ 238 °C, T_shoulder_ 260 °C, T_peak_ 322 °C, and T_offset_ 351 °C. The DTG curves reveal that during the decomposition of PR and EG, two different peaks, and therefore two degradation processes, corresponding to hemicellulose and cellulose were observed. The first peak (T_shoulder_) occurs at 309.25 for PR and 279.59 for EG. In addition, the DTG profile for EG showed a shoulder, confirming hemicellulose decomposition; in PR, this shoulder is not evident since the decomposition temperature ranges of hemicellulose and cellulose partially overlap [52]. The second peak (T_peak_) (345.61 °C for PR and 336.80 °C for EG) characterises the maximum decomposition rate of cellulose. The wide temperature range where lignin decomposes hinders the appearance of a peak attributable to this component. Also, volatile evolution at low temperatures is usually associated with extractive decomposition, but this shoulder is not evident [53]. The values of T_onset_ marked the starting point for the thermal decomposition of less thermally stable components. At 231.42 °C in PR and 229.41 °C in EG, volatiles were released from biomass species. T_offset_ marked the endpoint of active pyrolysis for PR (367.09 °C) and EG (357.23 °C).

The % Res becomes more significant as the heating rate increases because, at higher heating rates, the biomass reaches a specific temperature in a shorter residence time. Additionally, a shorter residence time due to an increased heating rate reduces the interaction period between biomass particles. It increases the volume of inorganic residue; whereas, at a slower heating rate, the extended residence time increases the interaction between biomass particles and decreases the volume of inorganic residue [54]. For the hardest wood of the tested samples, their thicker cell walls prevented heat transfer to the particle core, slowing thermal decomposition. This effect was reflected in the % Res value, which was 13.53% for PR and 20.61% for EG.

##### Activation Energy (E_a_) Using Isoconversional Models

Friedmann differential isoconversional, Flynn–Wall–Ozawa integral isoconversional, and Coats–Redfern model adjustment methods were used to determine the reaction’s E_a_ and evaluate the minimum energy required to initiate chemical reactions in the biomass. Appendix A show the thermogravimetric (TG), differential thermogravimetric (DTG), and second derivative of mass loss (D^2^TG) for PR and EG at 5, 10 and 20 °C·min^−1^, respectively. The curves are similar, indicating that the process’s general reaction mechanism is independent of the heating rates [55]. The curves are displaced to the right side as the heating rate increases from 5 to 20 °C min^−1^. This shift occurs due to the formation of thermal lag between biomass particles, as the interaction time between the particles decreases with the increase in heating rate. At this point, heat transfer is reduced, leading to the formation of thermal resistance between biomass particles. Consequently, a higher temperature is required for the biomass to initiate the devolatilisation process at a higher heating rate [54].

Table 4 shows the activation energy (E_a_), pre-exponential factor (A), and correlation coefficient (R^2^) for the raw biomass and two extreme pretreatment conditions. The relationship between the E_a_ and α values was calculated in a conversion range of 0.20 ≤ α ≤ 0.80. This conversion interval was chosen because it was more stable from the point of view of the pyrolysis reaction, and the exclusion of certain experimental data due to the lack of correlation will not affect the quality of the computed E_a_. As recommended by the ICTAC committee [56], since three heating rates were conducted in this work, the number of freedom degrees is 1; thus, in statistical terms, such a plot can be accepted as linear with 95% confidence only when its respective correlation coefficient R is more than 0.997 (R^2^ of 0.994) [57]. Therefore, the FWO and CR methods meet the ICTAC committee requirement, and the FR method does not. According to de Carvalho et al. [58], this result indicates that differential methods present inaccuracies due to numerical differentiation and become less precise than integral methods. The E_a_ values for PR and EG decreased as the irradiation power increased from the untreated to the 700 W-5 min condition, indicating that microwave pretreatment improves the components’ activity and enhances the structural compounds’ decomposition as reactions occur at a lower temperature with increasing irradiation, leading to a faster reaction rate.

Moreover, a lower E_a_ value signifies that the reaction occurs with a reduced external energy input [59]. A microwave electromagnetic field promotes molecular mobility within the biomass, elevating the pre-exponential factor (A). This factor is affected by the vibration frequency of the atoms in the biomass and intensifies with the application of microwaves [60]. The application of microwaves increased the reaction rate, as confirmed by the kinetic parameters. Dong and Xiong [61] conducted a kinetic study of moso bamboo pyrolysis, determining the E_a_ and heating rate when treated with microwaves (24.5 kJ· mol^−1^ and 160 °C·min^−1^) and compared this method with electric oven heating (58.3 kJ mol^−1^ and 135 °C·min^−1^). This decrease in activation energy suggests that microwave treatment would be a promising method before biomass pyrolysis. The range of R^2^ corresponding to all conditions ranged from 0.97 to 0.99, evidencing a high confidence in the fitting results.

#### 3.4.2. Fourier Transform Infrared (FTIR) Spectroscopy

Figure 3 shows the FTIR spectra of *Pinus radiata* and *Eucalyptus globulus*. The analysis of absorption bands was conducted in the wavelength range from 3600 cm^−1^ to 800 cm^−1^, as such a range exhibits the highest changes after MW pretreatment. Seeking clarity, the discussion focus on two sub-regions (i) from 3600 cm^−1^ to 2800 cm^−1^ and (ii) from 1800 cm^−1^ to 800 cm^−1^. The former includes O-H and C-H stretching bands at 3400 cm^−1^ and 2800 cm^−1^, respectively. The second is assigned to different stretching vibrations of other groups from cellulose, hemicellulose, and lignin in the range of 1800–800 cm^−1^ [62].

Regarding the O-H stretching region (3400 cm^−1^), the broad band observed for all of the EG samples resulted from variations in the degree of hydrogen bonding between hydroxyl groups in the cellulose crystal/aggregated states, hemicelluloses, lignin, and water [63]. This band weakened as microwave pretreatment was increased since -OH can dissociate in the form of dehydration from sugar cellulose rings. This dissociation phenomenon is facilitated as the temperature rises due to microwave irradiation [64].

Two bands at 2900–2800 cm^−1^ are composed by the overlapping of the stretch asymmetric vibrations of -CH_2_- (generally around 2935–2915 cm^−1^) and -CH_3_ (2970–2950 cm^−1^) and by the overlap of the stretch symmetric vibrations of -CH_2_- (2865–2845 cm^−1^) and -CH_3_ (2880–2860 cm^−1^). The apparent shift in frequency for the maximum CH band is due to structural and relative composition changes, namely changes at the cellulose crystallinity level, which influences the C-H and O-H stretch frequencies, and changes in the relative importance of lignin methoxyl groups for which the C-H stretching vibrations have lower CH stretching frequencies [65].

The band around 1740 cm^−1^ is assigned to the C=O stretching vibrations of the carboxyl and acetyl groups in hemicellulose. The intensity of this peak remains consistent following pretreatment, indicating a correlation with the hemicellulose composition achieved during microwave treatment [66]. The band at 1600 cm^−1^ is attributed to the aromatic skeletal vibration of lignin. It is related to the stretching vibration of the carbon–carbon double bond (C=C). The absorption intensity showed a weakened trend as the pretreatment temperature increased, suggesting a reduction in C=C bonds due to the increase in pretreatment intensity [64]. The absorption band around 1514 cm^−1^ is associated with the aromatic skeletal vibration (C=C) of the benzene ring in lignin. The decrease in the intensity of this signal is a consequence of the breakage of β-O-4 bonds in lignin as the microwave power increases [65]. The relative intensity of this band is higher in PR than in EG, due to its higher lignin content [67].

At the band of 1460 cm^−1^ (asymmetric C−H deformations in lignin), a slight decrease was noted. This change was caused by lignin degradation and the cleavage of methoxyl groups during microwave treatment [65]. The absorption band in the region of 1372 cm^−1^ is due to the angular deformation of C-H bonds (CH_2_) in the cellulose structure [68]. The intensity of these peaks decreases as microwave treatment increases, suggesting a reduction in cellulose content. The band at 1340 cm^−1^ is assigned to the guaiacyl ring, a structure in the lignin of both hardwoods and softwoods [69]. These peaks were intensified with the increase in microwave power. This demonstrated that microwave radiation improved the demethoxylation and cleavage of β-O-4 bonds in lignin, promoting lignin removal.

The band at 1278 cm^−1^ decreases in intensity and corresponds to C-O stretch vibration in hemicelluloses’ lignin, acetyl, and carboxylic vibration [70]. The bands are in the 1190 to 950 cm^−1^ area attributed to the C−O and C−H vibrations derived from aliphatic −CH_2_ or phenol −OH bonds. The slight decrease in absorbance at this region indicated the gradual degradation of methyl and hydroxyl groups.

The band at 1028 cm^−1^ (methoxyl groups in lignin) decreased permanently as the treatment intensified. This trend is attributable to the partial demethoxylation of lignin and its gradual crosslinking [68]. The peak at ~890 cm^−1^ is due to β-glycosidic linkages of the glucose ring within the cellulose structure [71,72].

The bands at 1430 cm^−1^ and 890 cm^−1^ represent the region of crystalline and amorphous cellulose (C-H deformation of cellulose) for PR and EG, respectively [65]. The signal intensity weakens as microwave irradiation increases due to cellulose dehydration, indicating that microwave pretreatment favours the reduction in cellulose crystallinity [73]. The crystallinity index (CI) estimated for the raw biomass and microwave-treated samples (Table 5) suggests that the pretreatment produces an internal fragmentation of the cellulose, which could result in an increment in the surface area. These effects could affect pyrolysis by reducing the bond-breaking energy requirements and facilitating the evacuation of volatiles, reducing the incidence of secondary reactions [13].

#### 3.4.3. Surface Morphology

Figure 4a–f shows the SEM images of the PR and EG samples (WOT, 259, and 700 W). The surfaces without microwave irradiation are in the form of compact blocks, and the structure is complete. However, a certain structure crack is observed when the biomass is irradiated with microwaves at 259 W-5 min. When the irradiation conditions increase in severity (700 W-5 min), the fragmentation of the structures is pronounced, even resulting in certain fragments detaching from the surface. Similar observations were reported by Liang et al. [74] in their study of microwave-treated pine sawdust pyrolysis under conditions of 126, 329, and 567 W for 3, 5, and 10 min. They believed that the high degree of perforation and distortion seen in pretreated samples is due to the microwave heating mechanism (from inside to outside), which converts electromagnetic energy into thermal energy at the molecular level. This result indicates the release of cellulose from lignin and hemicellulose in the biomass matrix [75]. It can remove the lignin layer from the interstices in the structure of packed microfibrils during pretreatment, causing interstice gaps.

Additionally, the non-thermal effect dominates the microwave heating process, forcing polarised side chains of large molecules in the cell wall to break their hydrogen bonds, altering the structure [74]. It is also observed that there is a greater disorder on the surface of EG treated with a 700 W microwave for 5 min (Figure 4f) compared to the surface of PR under the same treatment conditions (Figure 4c) because softwoods are more resistant to microwave irradiation than hardwoods. This is because EG has a higher cellulose content (53.0%) than PR (43.1%), and during the pretreatment, crystallinity is reduced, depolymerising cellulose by breaking hydrogen bonds and disrupting the crystalline arrangement of cellulose molecules [76].

#### 3.4.4. Extractives Characterisation

The variation in extractives content was significant from the untreated condition to the 700 W-5 min condition, as verified in Table 3. Then, biomass pyrolysis was carried out with and without extractives. Additionally, the extractive compounds were quantified through GC/MS to determine whether the variation in these compounds influences the formation of compounds during biomass pyrolysis. Table 6 shows the chemical compounds obtained from extractives determined by GC-MS. De Paula Protásio et al. [77] concluded that extractives’ acetone-soluble compounds represent chemical components such as fatty acids and resin acids, and their thermal degradation range is between 250 and 505 °C. Extractives are high-molecular-weight compounds that cannot be directly identified, so they must be derivatised [78]. The benzene carboxylic acid in extractives can be converted to esters, acids, and anhydrides. Each carboxyl group can react separately, so compounds can be prepared to alter the carboxyl groups into different derivatives [79]. For this reason, in the pyrolysis with extractives, the formation of acids compounds is more significant than that in the pyrolysis of biomass without extractives. Furthermore, compounds in the extractives appear to be released by simple evaporation during pyrolysis and contribute to the pyrolysis products (acetic acid, propanoic acid, oxalic acid). These compounds were also found in the study conducted by Mészáros et al. [80] on the thermal behaviour of *Robinia pseudoacacia* (black locust).

### 3.5. Py–GC/MS Analysis

#### 3.5.1. Family Compounds Formed during PR and EG Pyrolysis

Figure 5 shows the variations in pyrolysis compound families for PR and EG, with extractives (WE) and without extractives (WOE), for untreated conditions (WOT) and 259 and 700 W during 5 min of microwave irradiation.

The identified compound families constitute the typical compound families formed during pyrolysis: simple oxygenates (acids, esters, alcohols, ketones, aldehydes), phenols, and aromatic compounds, as reported by Wang et al. [81]. During microwave treatment, the cellulose contents of PR and EG were slightly reduced by 2% in PR and 3% in EG. Accordingly, a decrease in acid compounds was found for the pyrolysis vapours obtained from these samples compared to their untreated references (PR: 21.97% to 17.34% and EG: 27.72% to 24.13%). The acidic compound’s amount decreases for both PR and EG as microwave power increases due to cellulose’s thermal cracking caused by the microwaves’ passage [74]. This reduction in acidic compounds leads to secondary ketonisation reactions, resulting in a pyrolysis oil enriched in ketones (PR: from 17.39% to 19.59% and EG: from 16.68% to 20.26%). A similar trend was found by Liang et al. [74] in their pine sawdust pyrolysis tests pretreated with microwaves (567 W-10 min).

The content of low molecular weight acids and ketones could be related to the extractive content in the biomass. Therefore, pyrolysis assays were conducted for samples with (WE) and without extractives (WOE). Thus, comparing the pyrolysis results for WE and WOE samples, a decreasing trend in acid and phenol compounds confirmed the prior hypothesis. Additionally, an increase in the concentration of ketones and aldehydes was observed. These results are consistent with the study of the thermal behaviour of *Robinia pseudoacacia* (black locust) conducted by Mészáros et al. [80], who indicated that the amount of extractives present in the biomass does not influence the quality and quantity of the pyrolysis compounds formed since their decomposition begins at 120 °C. They decompose in the same temperature range as the other components of the biomass (hemicellulose, cellulose, and lignin). Thus, a decreased lignin content led to reduced phenolics (PR: from 34.41% to 31.95% and EG: from 21.73% to 20.24%).

#### 3.5.2. Chemical Compounds Formed during PR and EG Pyrolysis

Appendix A shows the most abundant chemical compounds in PR and EG pyrolysis after microwave pretreatment. Such results elucidated the pretreatment influence on the pyrolytic compound composition and the possible formation pathway (Figure 6).

The acetic acid yields are 10.85% for EG and only 6.72% for PR, which agrees with the xylan and high-acetyl group content of xylan in EG [80]. Autohydrolysis during the pretreatment results in the formation of acetic acid, which accelerates degradation. The decrease in acids is related to variations in lignin due to esterification caused by the reaction of the produced acids with the wood cell wall [82]. Hydroxyacetaldehyde is a compound that comes from hemicellulose, and this compound is mainly derived from the decomposition of xylan units, a component of hardwoods such as *Eucalyptus* [83]. Therefore, its presence is only in EG’s pyrolysis and not PR. Additionally, Usino et al. [84] indicated in their work on primary interactions of biomass components during fast pyrolysis that the main products derived from hemicellulose have a greater influence on products derived from cellulose. Among them are families of aldehydes, especially hydroxyacetaldehyde, ketones, and furans.

Furfural can be obtained from cellulose and hemicellulose pyrolysis and by lignin depolymerisation. Regarding composition, the hardwood hemicellulose is xylan, which has a greater propensity for furfural formation, favoured as microwave treatment intensity increases. Thus, the appearance of this compound is higher in EG (from 2.64% to 2.81%) compared to PR (from 1.75% to 2.21%) [66].

Moreover, the decrease in the lignin content led to phenolics’ reduction (PR: from 34.41% to 31.95% and EG: from 21.73% to 20.24%). Then, phenolic compounds’ (creosol, 2-methoxy-4-vinylphenol; phenol, 2-methoxy-4-(1-propenyl); phenol, 4-ethyl-2-methoxy; and phenol, 2,6-dimethoxy) reduction was associated with depolymerisation reactions and the cleavage of C-O and C-C bonds present in lignin because of the breakage of β-O-4 bonds in lignin as microwave power increases [43]. Phenol, 2,6-dimethoxy (syringol) is a phenolic compound formed by syringyl alcohol units in hardwoods such as *Eucalyptus*, which is why this compound is characteristic of EG pyrolysis [85].

#### 3.5.3. Reaction Pathways

Figure 6 shows the possible reaction pathway of the pyrolysis of PR and EG subjected to microwave irradiation pretreatment. The figure indicates whether a compound or family of compounds was generated in greater quantity in PR or EG, showing an increase or decrease in the compounds formed as the pretreatment condition varied.

In cellulose pyrolysis, a series of competitive depolymerisation reactions is generated at the end of the polymer chain and breaks the –β bond to form levoglucosan and acetic acid [86]. The absence of levoglucosan suggests that this intermediate first underwent a depolymerisation step. Then, from dehydration, fragmentation, and condensation reactions, it forms acids, furans, and aldehydes such as acetic acid, furfural, and methyl glyoxal [87]. Acetol is another product found during cellulose-fast pyrolysis, formed by the relocation of C-4 from the pyran ring cleft owing to levoglucosan degradation. Hydroxyacetaldehyde is obtained by the secondary degradation of levoglucosan or intermediate fragmentation during cellulose pyrolysis. A low amount of hydroxyacetaldehyde is produced as the result of secondary pyrolysis reactions [5], and its presence is only confirmed in pyrolysis products of EG (2.08% in EG-WE-WOT to 2.88% in EG-WE-700 W), due to the cellulose content in EG being higher than in PR.

**Figure 6 polymers-15-03790-f006:**
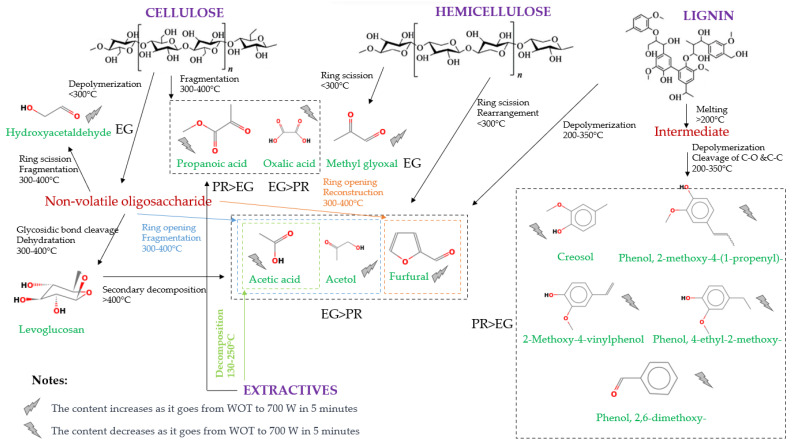
Proposed reaction scheme. Adapted from [5,88,89,90,91].

Hemicellulose pyrolysis comprises decomposition and polymerisation reactions, leading to the cleavage of glycosidic bonds, the formation of oligosaccharides, and further breakdown into xylose [5]. Acetic acid, methyl glyoxal, acetol, and furfural are the main compounds formed from xylan after a primary elimination reaction of the O-acetyl group attached to the main xylan chain at position C2 [92]. The higher yield of these compounds in EG pyrolysis than in PR pyrolysis is because xylan is the predominant polysaccharide in hardwood hemicellulose.

Lignin pyrolysis reactions start with a lignin softening, transforming lignin into several liquid intermediates. The second stage is the depolymerisation of these intermediates, turning them into aromatic alcohols known as monolignols, including coniferyl, sinapyl, and paracoumaryl alcohols. These organic compounds are then transformed into methoxyphenols, namely creosol, 2-methoxy-4-vinylphenol; phenol, 2-methoxy-4-(1-propenyl)-; and phenol, 4-ethyl-2-methoxy- by severing the C=O and C=C bond [91]. The formation pathway of 2-methoxy-4-vinylphenol during Type 4 lignin dimer decomposition into a-r1 and a-r2 radicals through Cβ-O bond cleavage was proposed by Yang et al. [93]. During cellulose pyrolysis, when microwave irradiation increases, the yields of these compounds decrease, both for PR and EG This occurs because the irradiation intensifies, and the microwaves rearrange the lignin [94]. This effect produced by microwave irradiation through the biomass also explains why the phenolic compounds mainly formed (creosol; 2-methoxy-4-vinylphenol; phenol, 2-methoxy; phenol, 2-methoxy-4-(1-propenyl)-; and phenol, 4-ethyl-2-methoxy-) reduced their yield as the microwave power increased.

## 4. Conclusions

In summary, this study evaluated the effects of microwave pretreatment on the analytical pyrolysis of *Pinus radiata* and *Eucalyptus globulus*. The most significant variations in maximum temperature (147.69 °C for PR and 130.71 °C for EG) and mass loss (11.06% for PR and 5.17% for EG) were obtained for the 700 W-5 min pretreatment.

Microwave pretreatment influenced the decrease in cellulose content due to the vibration and realignment of the (-OH) groups and the reduction in lignin content upon reaching its glass transition temperature. This lignocellulosic component’s rearrangement increased extractives from 1.8% to 7.2% in PR and from 1.9% to 6.5% in EG. As the microwave energy delivered increased, a decrease in activation energy and a pre-exponential factor increase was observed, indicating that microwave pretreatment enhances the activity of the components and improves the decomposition of structural compounds for subsequent thermochemical transformation processes such as pyrolysis. The surface fragmentation of biomass structures was confirmed after microwave heating.

The pyrolysis of pretreated biomass showed a reduction in oxygenated compounds: acids (from 21.97% to 17.34% for PR, and from 27.72% to 24.13% for EG) and phenols (from 34.41% to 31.95% for PR, and from 21.73% to 20.24% for EG), demonstrating a positive influence of microwave pretreatment, finally favouring properties of potential liquid pyrolysis products. Therefore, the microwave pretreatment of lignocellulosic biomass (PR and EG) is a promising technology for obtaining valuable compounds in pyrolytic processes.

## Figures and Tables

**Figure 1 polymers-15-03790-f001:**
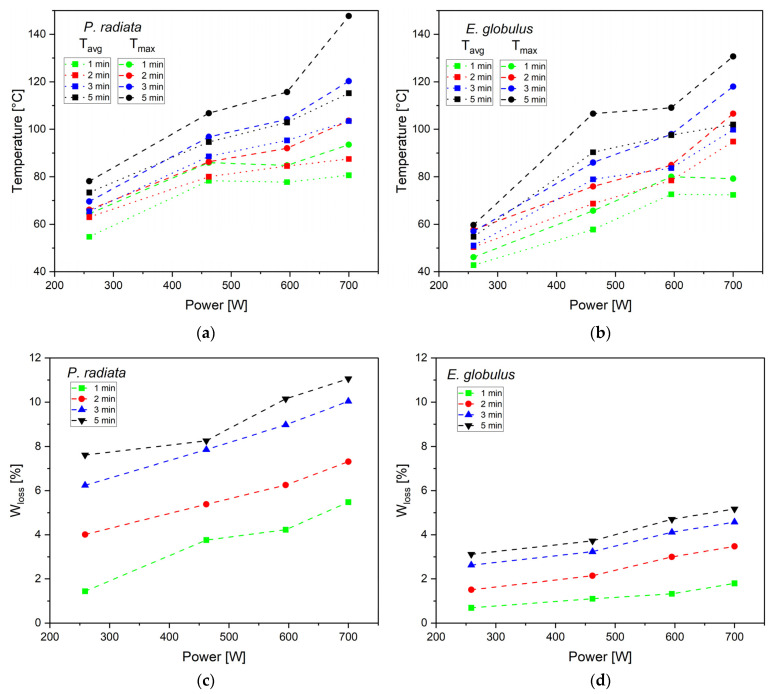
Average (T_avg_) and maximum (T_max_) temperatures reached by the biomass samples vs. power for (**a**) PR and (**b**) EG; and weight loss (W_loss_) vs. power for (**c**) PR and (**d**) EG.

**Figure 2 polymers-15-03790-f002:**
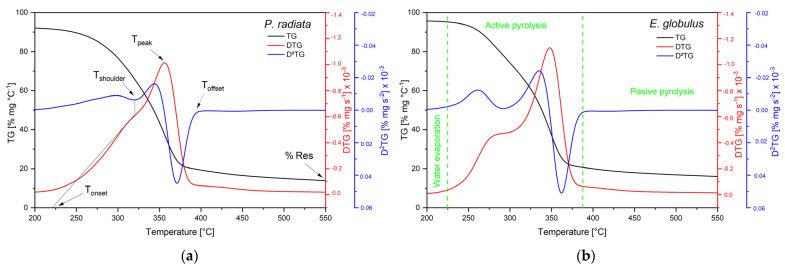
TG, DTG, and D^2^TG for (**a**) *Pinus radiata* (**b**) *Eucalyptus globulus*.

**Figure 3 polymers-15-03790-f003:**
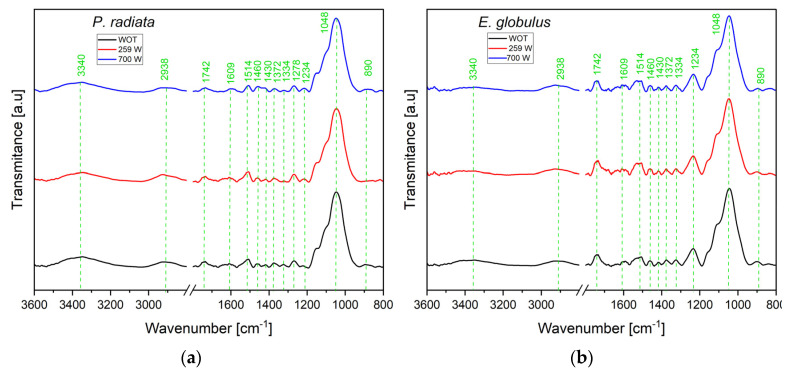
FTIR spectra of (**a**) *Pinus radiata* and (**b**) *Eucalyptus globulus* before and after pretreatment.

**Figure 4 polymers-15-03790-f004:**
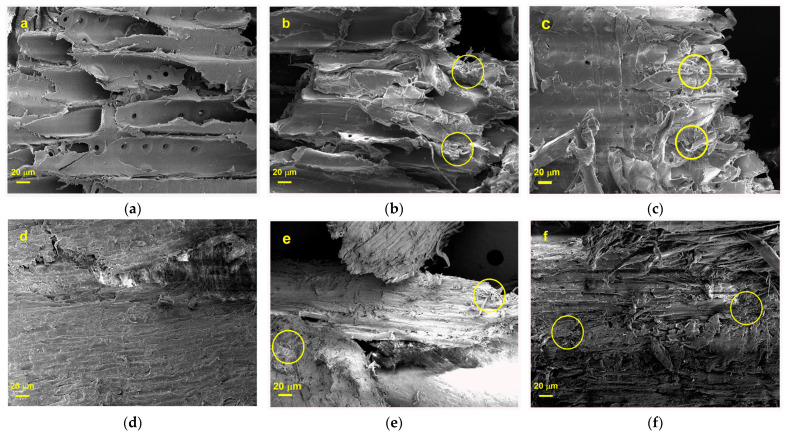
SEM images corresponding to: (**a**) PR WOT; (**b**) PR 259 W; (**c**) PR 700 W; (**d**) EG WOT; (**e**) EG 259 W; (**f**) EG 700 W.

**Figure 5 polymers-15-03790-f005:**
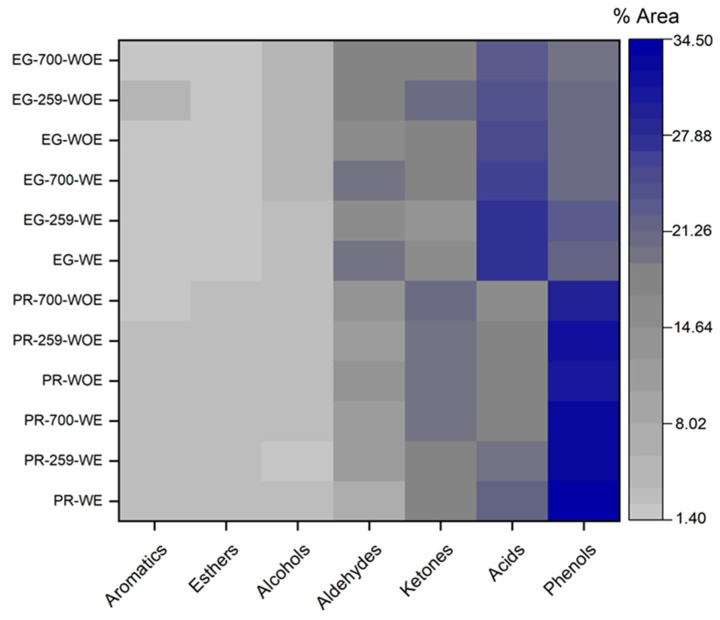
Pyrolysis family compounds for *Pinus radiata* with extractives (WE) and without extractives (WOE) and for *Eucalyptus globulus* with extractives (WE) and without extractives (WOE).

**Table 1 polymers-15-03790-t001:** Kinetic methods.

Method	Expression	Plots	Ref.
Friedmann	ln (dα/dt) = ln [A·f(α)] − E_a_/R·T	ln (dα/dt) vs. 1/T	[31]
Flynn-Wall-Ozawa	ln β = ln [A·E_a_/R·g(α)] − 5.3305 − 1.052 E_a_/R·T	ln β vs. 1/T	[32,33]
Coats-Redfern	ln [β/T^2^ (1 − 2R·T/E_a_))] = ln [−A·R/E_a_·ln(1 − α))] − E_a_/R·T	ln (β/T^2^) vs. 1/T	[34]

**Table 2 polymers-15-03790-t002:** Characterisation results of biomass samples.

	Proximate Analysis			Ultimate Analysis			Chemical Composition	
	PR	EG		PR	EG		PR	EG
Moisture (%)	7.75	5.51	Carbon (%)	48.02	47.76	Hemicellulose (%)	28.6 ± 0.8	26.3 ± 0.3
Volatiles (%)	76.73	77.13	Hydrogen (%)	5.90	6.32	Cellulose (%)	43.1 ± 0.1	53.0 ± 0.2
Fixed carbon (%)	14.68	16.85	Nitrogen (%)	0.29	0.09	Lignin (%)	26.6 ± 1.6	23.9 ± 2.1
Ash (%)	0.83	0.51	Sulphur (%)	0.10	0.05	Extractives (%)	1.8 ± 0.3	1.9 ± 0.0
HHV (MJ/kg)	18.98	19.38	Oxygen (%)	45.69	45.77			

**Table 3 polymers-15-03790-t003:** Chemical composition of PR and EG lignocellulosic biomass.

Samples	Pretreatment Condition	Hemicellulose (%)	Δ * (%)	Cellulose (%)	Δ * (%)	Lignin (%)	Δ * (%)	Extractives (%)	Δ * (%)
PR	259 W-5 min	29.3 ± 1.5	−0.7	41.0 ± 0.3	2.1	24.5 ± 1.3	2.1	3.0 ± 0.0	−1.2
	700 W-5 min	28.8 ± 0.7	−0.2	41.3 ± 0.4	1.8	24.4 ± 1.5	2.2	7.2 ± 0.1	−5.4
EG	259 W-5 min	28.0 ± 0.5	−1.7	50.0 ± 0.3	3.0	22.8 ± 1.7	1.1	3.8 ± 0.4	−1.9
	700 W-5 min	31.2 ± 0.2	−4.9	50.4 ± 0.2	2.6	22.3 ± 0.5	1.6	6.5 ± 0.2	−4.6

* Differences concerning the untreated biomass.

**Table 4 polymers-15-03790-t004:** Kinetic parameters and the characteristic temperatures from the thermogravimetric assays.

Biomass	Microwave Pretreatment	FR Method			FWO Method			Coats–Redfern Method		
		E_a_	A	R^2^	E_a_	A	R^2^	E_a_	A	R^2^
		kJ·mol^−1^	min^−1^		kJ·mol^−1^	min^−1^		kJ·mol^−1^	min^−1^	
	WOT	183.03	30.48	0.9898	201.71	39.23	0.9800	202.05	38.38	0.9780
PR	259	177.95	33.08	0.9956	185.07	40.83	0.9962	185.04	40.34	0.9967
	700	164.69	34.21	0.9973	174.91	45.01	0.9978	175.01	43.94	0.9971
	WOT	156.19	26.48	0.9956	174.80	34.28	0.9989	165.89	33.65	0.9990
EG	259	150.42	27.94	0.9911	166.72	35.94	0.9946	158.28	35.26	0.9938
	700	143.30	29.10	0.9889	158.51	37.57	0.9910	150.38	36.79	0.9898

**Table 5 polymers-15-03790-t005:** Crystallinity index (CI) for *Pinus radiata* and *Eucalyptus globulus* before and after treatments.

Sample	Pretreatment	Crystallinity Index
PR	WOT	1.31
	259-5	1.04
	700-5	0.82
EG	WOT	0.81
	259-5	0.68
	700-5	0.67

**Table 6 polymers-15-03790-t006:** Extractives compounds of PR and EG (WOT, 259–700 W).

Chemical Compounds	PR %Area			EG %Area		
	WOT	259	700	WOT	259	700
1,2-Benzenedicarboxylic acid	73.67	49.67	4.17	15.52	23.08	15.82
2-Propenoic acid	5.57	1.90	n.d	6.39	n.d	1.11
Acetic acid	4.18	8.47	2.27	43.05	4.02	24.14
Oxalic acid, allyl nonyl ester	2.09	11.29	45.02	4.70	15.72	20.54
Acetic acid, chloro-, methyl ester	n.d	0.00	16.55	12.40	6.48	6.45
Butanal, 3-methyl-	n.d	1.74	11.09	0.00	17.73	5.26
Acetaldehyde	n.d	2.48	12.97	n.d	n.d	10.63

*n.d*: non-detected.

## Data Availability

Data will be availaible by request.

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
