# Peer review of "Analytical Pyrolysis of Pinus radiata and Eucalyptus globulus: Effects of Microwave Pretreatment on Pyrolytic Vapours Composition"

_polymers, 2023, doi:10.3390/polym15183790_

Round 1
Reviewer 1 Report
In this paper, Analytical Pyrolysis of Pinus radiata and Eucalyptus globulus: Effects of microwave pre-treatment on pyrolytic vapours com- position. After careful evaluation of the manuscript, the work not acceptable. The author needs to have a serious attitude towards the article. There are some comments the author needs to clarify.
1. “Cellulose content decreased after pre-treatment from 43.1% to 41.3% in PR, and from 53.0% to 50.4% in EG,” A total of several parallel experiments were performed on the data to determine whether the reason for the decrease was within the error range.
2. It is recommended to annotate “% Res”.
3. It is suggested to unify the “Fig” and “Figure” in the article.
4. When using an acronym for the first time, the full form should be provided in parentheses to ensure clarity and comprehension. So the same abbreviation can be used when this term appears later in the text. For example,The full names of “WE” and “WOE” appear many times.
No
Author Response
Estimated Reviewer,
The authors appreciated your valuable comments, noting them and improving the article. Thus, we prepared a modified version of the original article to afford all comments.
Best Regards
Comments and responses
"Cellulose content decreased after pretreatment from 43.1% to 41.3% in PR, and from 53.0% to 50.4% in EG," A total of several parallel experiments were performed on the data to determine whether the reason for the decrease was within the error range.
R/ The original paragraph was modified, incorporating new information and enhancing comprehension of registered results. Lines 291-321.
It is recommended to annotate "% Res".
R/ Figure 2 was corrected.
It is suggested to unify the "Fig" and "Figure" in the article.
R/ The term Figure was chosen for all cases.
When using an acronym for the first time, the full form should be provided in parentheses to ensure clarity and comprehension. So the same abbreviation can be used when this term appears later in the text. For example, The full names of "WE" and "WOE" appear many times.
R/ Both acronyms were defined in lines 235-237
Reviewer 2 Report
The topic of the research work and manuscript is really interesting and provides new information. However there are some issues to be addressed towards its quality improvement before publication. The abstract is well-prepared though it should be shortened according to the intructions to authors of this journal. More and simpler words should be added in key words in order to contribute to the readability of your manuscript (avoid phrases). In lines 48-49, grammatical correction (the transitioning is imperative). In line 54, please add as well the relevant study of https://doi.org/10.3390/su132212810 . Could the referred "decomposition of lignin due to reaching its glass transition temperature" be reached at the level temperature of 147.69 °C (Pinus) or at 130.71 °C (Eucalyptus) for such biomass?You could provide some references on that, since the plasticization of lignin usually begins at higher that this temperature range, presenting in parallel further cross-linking and increase of lignin network, not the decomposition of lignin (based on the so far described response of wood chemical components to heat). The formation of new extractives usually comes from the hemicelluloses and amorphus parts of cellulose decomposition at such low temperature range. Lines 34-35 need clarification. The term "hemicellulose" would rather be used in plural form due to the nature of the component. In line 61, please add the word "biological" in front of the "resistance". In line 124, not the hemicelluloses? Was in the chips of the species used as raw material (PR and EG) came from sapwood/heartwood or both?since the chemical composition of them differs. Did you use wood/bark or both? did you use more than one trunk for the material of each species examined? Which was the moisture content of the raw materials before use? In 135 line, the "untreated" should come before the "Biomass". In microwave irradiation, please describe the specimen characteristics. Reference for equation 5? In 206 line, the "raw and pre-treated" is not very clear. Leave space between the value and celsius unit symbol. Please, provide as well (where possible and where is missing) the standard deviation values, except for the mean values in tables. Species scientific names in italics in the whole text (check conclusions for example).
Adequately satisfying use of English language. Only minor improvement in specific points in the text.
Author Response
Estimated Reviewer,
The authors appreciated your valuable comments, noting them and improving the article. Thus, we prepared a modified version of the original article to afford all comments.
Best Regards
Comments and responses
The abstract is well-prepared though it should be shortened according to the intructions to authors of this journal.
R/ The abstract was modified to:
Pinus radiata (PR) and Eucalyptus globulus (EG) are the most planted species in Chile. This research aims to evaluate the pyrolysis behaviour of PR and EG from the Bio-Bio region in Chile. Biomass samples were subjected to microwave pretreatment considering power (259, 462, 595, and 700 W) and time (1, 2, 3, and 5 min). The maximum temperature reached was 147.69 °C for PR and 130.71 °C for EG in the 700 W - 5 min condition, which caused the rearrangement of the cellulose crystalline chains through vibration and an increase in the internal energy of the biomass and the decomposition of lignin due to reaching its glass transition temperature. Thermogravimetric analysis revealed an activation energy (Ea)reduction from 201.71 to 174.91 kJ·mol-1 in PR and from 174.80 to 158.51 kJ·mol-1 in EG, from untreated condition (WOT) to the 700 W - 5 min condition, which indicates that microwave pretreatment improves the activity of the components and the decomposition of structural compounds for subsequent pyrolysis. Functional groups were identified by Fourier transform infrared spectroscopy (FTIR). A decrease in oxygenated compounds such as acids (from 21.97 to 17.34 %w·w-1 and from 27.72 to 24.13 %w·w-1) and phenols (from 34.41 to 31.95 %w·w-1 and from 21.73 to 20.24 %w·w-1) in PR and EG, respectively, was observed from WOT to 700 W - 5 min condition, after analytical pyrolysis. Such results demonstrated the positive influence of the pretreatment on the reduction of oxygenated compounds obtained from biomass pyrolysis.
More and simpler words should be added in key words in order to contribute to the readability of your manuscript (avoid phrases).
R/ Keywords were modified to: Biomass, pyrolysis, pretreatment, microwave, oxygenated compounds.
In lines 48-49, grammatical correction (the transitioning is imperative).
R/ The text was corrected to: Given this situation, transitioning from this fossil-based economic model to alternative bio-based economies is imperative.
In line 54, please add as well the relevant study of https://doi.org/10.3390/su132212810
R/ The reference was incorporated.
Could the referred "decomposition of lignin due to reaching its glass transition temperature" be reached at the level temperature of 147.69 °C (Pinus) or at 130.71 °C (Eucalyptus) for such biomass?.
R/ The redaction was enhanced, as can be revised in lines 291-321 of the article new version:
Table 3 shows the variation of the lignocellulosic species' chemical composition after the microwave treatment. In all cases, the difference between the standard deviation and the arithmetic mean is less than 6%, except for the extractives in the 700 W - 5 min condition, which reaches 10.52%. Such variations in chemical components are due to the pretreatment effect and the microwaves acting inside the biomass and cannot be significantly attributed to experimental measurement error. The interactions between electromagnetic waves and matter are expressed as conductive losses, dielectric losses and magnetic losses [39]. The alignment of polar molecules caused by microwave irradiation forces the dipoles of the molecules to align in the radiation field (polarised). The rearrangement of the polar molecules produces a displacement within the material that generates heat [40].
In the maximum condition (700 W - 5 min), the cellulose content decreased after pretreatment from 43.1% to 41.3% in PR and from 53.0% to 50.4% in EG. Such variation can be associated with microwave irradiation that leads to intense molecular collisions due to dielectric polarisation, potentially decomposing the cellulose content [41]. Then, the cellulose structure's hydroxyl groups (-OH) make it a polar molecule. Then, as the microwaves pass through these polar molecules, they realign and vibrate them, increasing their internal temperature and displacement [18]. In this condition, lignin content decreased after pretreatment, from 26.6% to 24.4% in PR and from 23.9% to 22.3% in EG. Knowing that the glass transition temperature of lignin (130-200°C) is usually lower than that of cellulose (230-250°C) and close to hemicellulose (160-200°C) under dry state [42], therefore, the decrease in this component may be associated with the initiation of decomposition due to reaching its glass transition temperature, which results in a disorder of the lignin structure.
Additionally, under these circumstances, new extractives can be generated when the wood is thermally treated due to the degradation of other components, such as lignin [43,44]. This generation of new extractives is reflected in its increase from 1.8% to 7.2% in PR and from 1.9% to 6.7% in EG. Finally, the arousal effect of the molecules by microwave application only increases the internal kinetic energy of the biomass, and since microwaves have low energy in their photons (0.03 kcal/mol), they do not directly affect the molecular structures of the biomass, as their chemical bonds have an energy ranging from 20 to 80 kcal/mol, resulting in a general percentage increase in hemicellulose content [45].
You could provide some references on that, since the plasticization of lignin usually begins at higher that this temperature range, presenting in parallel further cross-linking and increase of lignin network, not the decomposition of lignin (based on the so far described response of wood chemical components to heat).
R/ Incorporated into previous response, Lines 291-321.
The formation of new extractives usually comes from the hemicelluloses and amorphus parts of cellulose decomposition at such low temperature range.
R/ Incorporated into previous response, Lines 291-321.
Lines 34-35 need clarification. The term "hemicellulose" would rather be used in plural form due to the nature of the component.
R/ Authors can't find the term in the commented lines. However, authors will consider such recommendations in the future.
In line 61, please add the word "biological" in front of the "resistance".
R/ Line 54 incorporated this recommendation.
In line 124, not the hemicelluloses?
R/ Corrected in lines 116-118.
Was in the chips of the species used as raw material (PR and EG) came from sapwood/heartwood or both?since the chemical composition of them differs. Did you use wood/bark or both? did you use more than one trunk for the material of each species examined? Which was the moisture content of the raw materials before use?
R/ Lines 124-126 now incorporate the following sentence: The chips were exclusively made of wood without bark, and the position of the sample within the trunk was not considered (sapwood/heartwood).
In 135 line, the "untreated" should come before the "Biomass". In microwave irradiation, please describe the specimen characteristics.
R/ Biomass was described in the raw material section.
Reference for equation 5?
R/ Incorporated in line 191
In 206 line, the "raw and pre-treated" is not very clear.
R/ The sentence was modified to: Mid-infrared spectra of untreated and pre-treated biomass samples were collected using a Thermo Spectrometer Scientific Infrared Nicolet is10 with DTGS detector, equipped with a Quest ATR accessory (Specac, UK).
Leave space between the value and celsius unit symbol.
R/ Thanks for pointing out this. The temperature units were corrected in the manuscript's new version.
Please, provide as well (where possible and where is missing) the standard deviation values, except for the mean values in tables.
R/ Such values were incorporated in Supplementary information.
Species scientific names in italics in the whole text (check conclusions for example).
R/ Scientific names were written in Italics.
Round 2
Reviewer 2 Report
As I have checked the authors have implemented the proposed changes in the revised verion of manuscript towards the improvement of their work. Almost all the changes have been implemented and in my opinion, the manuscript is well-prepared and organized enough to be accepted for publication in this journal.